# Myelin Quantification in White Matter Pathology of Progressive Multiple Sclerosis Post-Mortem Brain Samples: A New Approach for Quantifying Remyelination

**DOI:** 10.3390/ijms222312634

**Published:** 2021-11-23

**Authors:** Marije J. D. Huitema, Eva M. M. Strijbis, Antonio Luchicchi, John G. J. M. Bol, Jason R. Plemel, Jeroen J. G. Geurts, Geert J. Schenk

**Affiliations:** 1Department of Anatomy and Neurosciences, Amsterdam Neuroscience, MS Center Amsterdam, Amsterdam UMC, VU University Medical Center, De Boelelaan 1108, 1081 HV Amsterdam, The Netherlands; marijejd@gmail.com (M.J.D.H.); a.luchicchi@amsterdamumc.nl (A.L.); jgjm.bol@amsterdamumc.nl (J.G.J.M.B.); j.geurts@amsterdamumc.nl (J.J.G.G.); 2Department of Neurology, MS Center Amsterdam, Amsterdam UMC, VU University Medical Center, 1081 HZ Amsterdam, The Netherlands; e.strijbis@amsterdamumc.nl; 3Department of Medicine, Division of Neurology, Neuroscience and Mental Health Institute, University of Alberta, Edmonton, AB T6G 2S2, Canada; jrplemel@ualberta.ca; 4Department of Medical Microbiology & Immunology, University of Alberta, Edmonton, AB T6G 2S2, Canada

**Keywords:** multiple sclerosis, remyelination, demyelination, myelin quantification, repair

## Abstract

Multiple sclerosis (MS) is a demyelinating and neurodegenerative disease of the central nervous system (CNS). Repair through remyelination can be extensive, but quantification of remyelination remains challenging. To date, no method for standardized digital quantification of remyelination of MS lesions exists. This methodological study aims to present and validate a novel standardized method for myelin quantification in progressive MS brains to study myelin content more precisely. Fifty-five MS lesions in 32 tissue blocks from 14 progressive MS cases and five tissue blocks from 5 non-neurological controls were sampled. MS lesions were selected by macroscopic investigation of WM by standard histopathological methods. Tissue sections were stained for myelin with luxol fast blue (LFB) and histological assessment of de- or remyelination was performed by light microscopy. The myelin quantity was estimated with a novel myelin quantification method (MQM) in ImageJ. Three independent raters applied the MQM and the inter-rater reliability was calculated. We extended the method to diffusely appearing white matter (DAWM) and encephalitis to test potential wider applicability of the method. Inter-rater agreement was excellent (ICC = 0.96) and there was a high reliability with a lower- and upper limit of agreement up to −5.93% to 18.43% variation in myelin quantity. This study builds on the established concepts of histopathological semi-quantitative assessment of myelin and adds a novel, reliable and accurate quantitative measurement tool for the assessment of myelination in human post-mortem samples.

## 1. Introduction

MS is a heterogeneous and complex demyelinating and neurodegenerative disease of the central nervous system (CNS). MS is characterized by extensive white and gray matter demyelination which is profoundly present as focal lesions [1,2,3]. Demyelination can be naturally followed by remyelination, but this is highly variable between people with MS [4,5]. It occurs in all lesion types in both gray and white matter [6,7] and may underlie clinical remission by restoring damaged myelin sheaths. Several hypotheses exist for why remyelination often stagnates [1,5,6,8,9,10]. Inhibition of oligodendrocyte recruitment to lesions, failure of oligodendrocyte precursor cell differentiation and maturation, or both, might underlie decreased remyelination capacity [6,7,9,11] CNS-resident microglia and monocyte-derived macrophages regulate the remyelination process by clearing inhibitory myelin debris and stimulating oligodendrocyte differentiation by shifting towards an immunoregulatory state [12,13,14,15,16]. However, it is still unclear whether microglia/macrophages can also arrest remyelination [7,12,16,17]. This highlights the importance of the lesion environment in regulating remyelination for people with MS [10,18].

To improve our understanding of remyelination it is crucial to both identify and locate it accurately. Throughout the years, multiple histological classification criteria for remyelination have been used. Paler Luxol-fast blue (LFB) stained areas are indicative of the thinner and shorter myelin fibers that are associated with remyelination [19,20]. In lesions, remyelination can occur partially or (almost) completely [1,19,20,21,22]. Partially remyelinated lesions are characterized by a thinner myelin LFB gradient restricted to the lesion rim or thin patchy myelin fibers in lesions. Lesions that are almost completely remyelinated (>60%) are named shadow lesions (SLs) and account for 20% of all WM lesions [19,20,22]. They show an almost normal myelin thickness and are therefore difficult to separate from normal appearing white matter (NAWM) [1,3,19]. In contrast to focal lesions, ‘dirty appearing’ or ‘diffusely abnormal’ white matter (DAWM) is characterized by its intermediate and more diffuse appearance of myelin and it is still unclear whether it represents normal, re- or demyelinated areas [23,24]. Roughly 40% of WM lesions show signs of remyelination. However, in previous studies, the extent of remyelination is often a rough estimation due to the lack of a standardized myelin quantification tool. Currently, the most widely used manner to quantify remyelination in MS lesions is by using a semi-quantitative scoring method. This method consists of four crude scoring gradations with each having a potential difference of approximately 40% remyelination within a scoring category [19,20,22].

Here, we describe a new tool called the myelin quantification method (MQM) that obtains more precise estimates of myelination densities. This allows us to quantitatively compare microstructural myelin changes in demyelinated or remyelinated lesions but importantly also in NAWM, DAWM and healthy controls. Application of the MQM may help to reveal the determinants of remyelination in more detail. So far, no method has yet been developed for objective quantification of myelin. By combining this novel method with previously established semi-quantitative methods, we found that the degree of myelination can be determined more precisely in post-mortem brain tissues.

## 2. Results

Three raters independently assessed 55 WM lesions independently using the myelin quantification method (MQM), resulting in a total of 55 assessments per rater. Based on these assessments, rater 1 found a mean of 31.47% black pixels (±27.04% [SD]), rater two found a mean of 32.67% black pixels (±28.85% [SD]), and rater three found 32.49% black pixels (±24.95% [SD]).

### 2.1. The Reliability of the Myelin Quantitative Measurements

For the MQM, excellent reliability was observed, with an ICC value for myelin quantification of 0.96 (95% CI [0.94–0.98]).

Bland-Altman plots for inter-rater reliability provided graphical information about the difference in myelin quantity between raters (Figure 1A–C), as well as any bias that may be present. Despite slightly different threshold sets between raters, the Bland-Altman plots indicate a mean difference of 4.50% (±4.26 [SD]) myelin, with a 95% limit of agreement of −3.85 to 12.85% myelin; bland-Altman plot 1A (rater 1 and rater 2). Plot 1B (rater 1 and rater 3) showed a bias of 5.53% myelin (±5.47 [SD]), with a 95% limit of agreement of −5.18 to 16.25% myelin. Plot 1C (rater 2 and rater 3) yielded a mean difference of 6.25% myelin (±6.21 [SD]), with a 95% limit of agreement of −5.93 to 18.43% myelin.

### 2.2. Reliability of Myelin Quantification Using ImageJ Built-In Automatic Global Threshold Calculation

To assess if our method for myelin quantification can be further standardized and automated, the built-in automatic threshold option in ImageJ (‘Default’) was applied to all 55 WMLs and the myelin quantity was obtained. The ICC was recalculated by adding the automatically generated results with the ‘default’ method as the fourth observer (computer). This resulted in a good reliability, with an ICC of 0.84 (95% CI [0.634–0.919]). These automatically generated outcomes of the ‘default’ built-in method were compared with the outcomes in myelin quantity of rater 1, because this rater was the most experienced. A mean difference of 14.29% myelin, with a 95% limit of agreement of −9.29 to 37.87% myelin was found for the ‘default’ method and rater 1 (Figure 1D). Log transformations were applied when indicated and did not specify proportional bias.

### 2.3. Quality Analysis—Myelin Computation with Different NAWM Magnifications

To understand if optical magnification affected the MQM, we performed pixel computation in two lesions: Lesion 1 was a demyelinated lesion and lesion 2 was a remyelinated lesion. We computed lesion 1 using MQM by comparing it to 3 NAWM areas, all with different magnifications (Appendix A). We find that both low and high magnification of these NAWM ROIs provided that same myelin quantity value of 0.44%. We also computed the myelin quantity in lesion 2 using the same 3 NAWM areas taken with consistent magnification. We find the same value of 49.68% regardless of magnification. This quality analysis shows there is no difference in the outcome of the estimated percentage myelin in lesions when NAWM parameters are altered, indicating that the magnification of the images is independent of the myelin density measurements.

### 2.4. Quality Analysis—Black Pixel Ratio in Lesions with Different Image Magnifications

Another quality analysis shows the differences in black (myelin) pixel ratios of two lesion images taken at different magnifications, when applying the same threshold value (Table 1, Figure 2). With lower magnifications, the analysis had fewer total pixels within the lesion, but this difference in pixel counts did not alter the black pixel ratio. Thus, the different magnifications of the lesion images used are comparable with one another. For data analyses the highest lesion magnification was used.

### 2.5. Non-Neurological Control

The myelin quantity in the NAWM of MS patients was compared to the normal white matter of non-neurological control (NNCs) to study if there were any differences in myelin density. Five brain samples of NNCs were included and the myelin quantity was measured in three different normal WM (NWM) ROIs (Figure 3A,B). There was no statistically significant difference in the myelin quantity between WM of NNCs (mean 96.94%, SD 1.91%) and NAWM of MS patients (mean 96.88%, SD 1.71%; *p* = 0.949, Figure 3C).

### 2.6. Sub Analyses—Myelin Swellings, DAWM and Encephalitis

Using the LFB staining we observed an interesting phenomenon that confirms our recent observations [25]. Swellings within the myelin structure were plentiful and most frequently observed in the proximity of demyelinated or still actively demyelinating lesions (Appendix A). These swellings were also seen in remyelinated lesions, however less frequently and to a lesser extent. They are easily identifiable using the LFB stain (Appendix A) and may serve as an additional clue for the absence or presence of remyelination. Sub analyses were performed to show the method’s efficacy for WM myelin assessment in conditions other than MS remyelination. NAWM and DAWM of MS patients and normal white matter (NWM) of NNCs were inspected (Appendix A), as well as an encephalitis case (Appendix A).

## 3. Discussion

We here report on a new histopathological myelination quantification method, which results in more precise estimates of myelin densities compared to the previously used semi-quantitative scoring systems, while building on those established methods to take the morphological state of myelin into account [19,20,22]. The semi-quantitative methods consist of a scoring system with wide ranges that is susceptible to scorer bias [19,20,22]. The purpose of this work is to develop and validate a myelination scoring method to better estimate remyelination in MS post-mortem brain tissue. We assessed and found high levels of inter-rater reliability between three independent raters.

Three raters used MQM in 55 WMLs in this study, which resulted in an excellent inter-rater reliability (0.96). Using Bland-Altman plots we defined the limits representing maximum differences in measurements between and found agreement were up to −5.93% to 18.43% the difference in myelin presence. At the extreme, the 18.43% interrater difference is still more precise than using semi-quantitative methods (with observer discrepancy ranges of up to 40%) [19,20,22]. However, using MQM we are still dependent on the visual histological inspection of the myelin structure (using the remyelination criteria) to distinguish between demyelination and remyelination. The Bland-Altman plots show that the majority of the differences in measurements were within 1.96 SD of the mean difference, which substantiates the inter-rater agreement [26,27]. When higher myelin percentages were present within lesions, the difference between raters diminished. Conversely, setting a representative threshold may become increasingly difficult with smaller percentages of myelin present (Figure 1). Lesions devoid of myelin are often heavily contrasted, because the transition from their completely demyelinated (white) center to the myelinated (black) borders is instant, without an LFB-stained gradient. As a result, many different threshold values when applied will yield similar binary images. Investigating completely demyelinated lesions thus increases the probability of raters choosing very dissimilar threshold values with corresponding myelin density outcome variations between raters. This is a potential limitation of our approach that needs to be taken into account; at the same time, fully demyelinated lesions are often well-recognized by routine light microscopic inspection in the first place.

### 3.1. Non-Neurological Controls, DAWM, Encephalitis

We demonstrated that the MQM may be useful in other brain histopathology than classical MS lesions. In the NNC’s there was statistically no difference in myelin density of normal white matter (NWM) compared to NAWM density in MS patients. NWM in NNCs can therefore be used as a baseline for normal myelin densities. Applying the method to NNCs gives the opportunity to study other abnormalities or diseases that display intermediate myelin density between those of focal lesions and NWM/NAWM, which could, until now, only be examined by histopathology and microscope [28]. For instance, we tested the quantification method on DAWM in MS patients. DAWM is characterized by its intermediate and more diffuse myelin appearance. Whether DAWM represents new demyelinating areas or represents a more chronic state of white matter integrity loss, or is just an expression of normal physiological anatomical variation, is still debatable [29]. DAWM may contribute to disease progression due to ongoing axonal pathology and inflammation [23,29]. Other studies define DAWM as an early manifestation of only microscopically visible WM lesions [28,30,31]. Applying the MQM to an example of DAWM, we were able to detect a small reduction in myelin density (by 6.63%) compared to the mean value of the myelin density of three corresponding NAWM regions. Although DAWM lacks a well demarcated border, using MQM we were able to better define the diffuse area. Gaining more knowledge about myelin densities combined with other cellular markers (e.g., axons, microglia and oligodendrocytes) in these diffuse areas may help us to further understand the underlying pathology of DAWM and lesion formation and recovery.

Using myelin density measures of normal myelin as control values, we used MQM to quantify an encephalitis case and found that this technique is suitable to see myelin loss following brain inflammation (71.3%; Appendix A). Demyelination is known to occur in encephalitis [32,33]. Therefore, MQM is likely relevant for investigating other myelin-related diseases. To note, we only validated the method for images that were originally stained with LFB, although it may be applicable to immunohistochemical stains using myelin antibodies as well [34].

### 3.2. Study Strengths and Limitations

One limitation we encountered with MQM is determining the appropriate threshold values for a given lesion. The goal of setting a threshold is to choose a cutoff point that produces a black and white image that resembles the original histological image as much as possible. Hence, choosing the appropriate threshold is susceptible to observer bias. An alternative to variable thresholding is to use the same threshold value for every image. However, this approach could yield under- and overestimations because of LFB staining variability and differences in day-to-day image collection. Still, using MQM with variable thresholding, we find high inter-rater reliability, suggesting that accounting for staining and microscopy variability is in fact possible. A variable threshold also provides a more accurate myelin estimation within a lesion because it draws from expert opinion. With manual staining of fixed tissue sections, it is difficult to uniformly apply histological staining, which is an inherent disadvantage of all staining methods in post-mortem research [35]. Established markers for remyelination are hard to find, although Fard et al., report on BCAS1 as a marker for early myelinating oligodendrocytes that are found in a proportion of chronic white matter lesions of patients with MS [36].

Another limitation we could not overcome, is that the location of the ROI contributes to the estimated values of myelin, since it is drawn manually and thus susceptible to observer bias. However, using NAWM as standard maximum myelin density based on three ROIs has limited this issue.

Lastly, we here assume that myelin loss is caused by a destructive process and that remeyelination by oligodendrocytes, enwrapping the axon initially as a thin layer of myelin, results in demyelination looking quite different from remyelination. This leaves open the possibility that during demyelination myelin sheets are retracted in a subtle and orchestrated manner by the oligodendrocyte, making them look similar to the thin layer of myelin typically observed with remyelination. Although formation of myelin blisters is enriched in the MS brain [25], it is currently unknown if subtle retraction of myelin happens at all in MS. Nevertheless, this is a limitation of our study, since we cannot distinguish these two processes. In a similar vein, using our approach it is impossible to distinguish late stage, complete remyelination from normal myelination. It is important to keep these limitations in mind when interpreting MQM data.

The flow chart, which systematically describes how lesions should be included and summarizes the characteristics of remyelination and demyelination (Figure 4), makes a more objective microscopic assessment of lesions possible, as it was very difficult to distinguish between remyelinated lesions and lesions with remyelination that underwent demyelination again. In addition, the LFB staining showed myelin swellings. These were most frequently observed in demyelinated lesions (Appendix A). These swellings are probably a first sign of lesion formation [25,37]. These swellings are thought to be caused by a disturbance of the axon-myelin interaction, through alterations in adhesion molecules, which are engaged in holding axons and myelin together [25,38]. Although we did see swellings in remyelinated lesions, this can still be due to demyelination, as remyelinated lesions contain less firm myelin and are more susceptible to another demyelinating event [39]. In general, remyelination criteria are hard to establish with great precision and objectivity. Classification criteria, as described in previous studies, were diverse concerning remyelination and lacked concrete description of demyelination features. Myelin swellings may be informative for distinguishing de- and remyelination. However, quantitative research is warranted to study this observation further and to explore if myelin swellings could then be added to the demyelinating features described in the flow chart of the current study.

Thus, combined with classical pathological inspection by eye, the MQM provides a more precise estimate than methods used in previous studies [19,20,22]. Outcomes provided by the current method may fit better with the actual extent of remyelination that occurs in progressive MS patients due to its smaller variance in estimations and the use of a continuous scale and a standardized computerized system. The estimate of myelin quantity can now be determined automatically instead of via subjective visual semi-quantitative determination only.

### 3.3. Conclusion and Future Prospects

This study introduces an improved quantitative measurement tool for remyelination in post-mortem MS tissue and examines its reliability. By combining this novel quantification method in ImageJ with the microscopically examined remyelination criteria, the degree of remyelination can be estimated more precisely and objectively in post-mortem brain tissue of MS patients. We find a mean difference between raters of approximately 5% and an excellent inter-rater agreement. In the future, we expect this quantitative method to complement the semiquantitative methods by its more accurate estimation of myelin and its applicability to other MS pathology, other human diseases and animal models. Subsequent studies are needed to discover whether this method provides useful information of clinically relevant pathology. A potentially fruitful approach is combining (post mortem) imaging methods with the MQM [40]. Finally, to limit observer bias and therefore provide more objective myelin estimates, ImageJ built-in methods for an automatic global threshold calculation may be useful. Further research is warranted to improve global automated threshold algorithms applicable for the quantification of myelin in brain tissue.

## 4. Materials and Methods

### 4.1. Sample Collection

Paraffin-embedded post-mortem MS brain tissue was obtained in collaboration with the Netherlands Brain Bank (https://www.brainbank.nl/, accessed on 13 February 2020), located in Amsterdam, The Netherlands. Because this study was part of an ongoing effort, we were able to acquire brain tissue which was monitored and selected prospectively using MRI guided autopsies, as this is the most accurate way of acquiring lesion samples [29]. Tissue blocks were fixed in formalin and embedded in paraffin. Tissue blocks were sliced in 10 µm thick sections and fixated on positively charged Superfrost Plus glass slides (VWR international). MS cases were selected when WM pathology was present in plaque panels of PLP-stained sections. Anatomical exclusion criteria were tissue sections from cerebellar-, brainstem- hippocampus and spinal cord regions. For the assessment of myelin quantity, cross-sectional tissue samples from 14 MS patients (five male, nine female) and five age-matched, non-neurological controls (NNCs) (two male, three female) were histologically stained with Luxol Fast Blue (LFB MBS; Gurr #16765, Electron Microscopy Sciences, Hatfield, PA, USA). The data for this study were obtained by quantifying myelin in WM lesions. Subanalyses were performed to show the method’s efficacy for NAWM and DAWM of MS patients and normal white matter (NWM) of NNCs (Appendix A). Demographic variables and study outcomes (e.g., age at diagnosis and mean remyelination percentage) are summarized in Appendix A. All tissue was collected with full informed consent for autopsy, the use of clinical information and physical material for research purposes from the donor or their next of kin.

### 4.2. Histological Processing

#### 4.2.1. Luxol Fast Blue

To demonstrate shadow lesions or partially remyelinated lesions and their remyelinating properties, staining of the lipoproteins of myelin sheaths was performed with LFB (Gurr #16765, Electron Microscopy Sciences, Hatfield, PA, USA). Sections were first deparaffinized by softening the paraffin on a heating plate (58 °C for 30–60 min) and then in a series of 100% xylene (3 × 10 min), 2 × 5 min 100% ethanol and 5 min in 96% ethanol. Subsequently, tissue sections were incubated in a 0.1% LFB solution in a stove at 58 °C overnight. All sections were washed and differentiated one by one. Washing steps were performed consecutively in 96% alcohol (2–3 s) and milli-Q (MQ) water (3 s). Immediately thereafter, differentiation steps were performed in 0.05% Lithium carbonate solution (Merck Millipore; 554-13-2; 5 s) and 70% ethanol (5–7 s) until decoloring of the cortical GM, whilst WM remained blue. The latter step was performed carefully and was checked microscopically to ensure homogeneity of the LFB staining. The differentiation steps were repeated if needed. The sections were then rinsed in MQ water. Finally, the samples were dehydrated in a series of ethanol 96% (3–5 min), 100% (2 × 5 min) and xylene (3 × 5 min). The sections were mounted with entellan and a coverslip.

#### 4.2.2. PLP Staining

All sections were immunohistochemically stained for proteolipid protein (PLP) to distinguish the WM-GM border and to identify areas of demyelination. These sections were stained in different batches, spanning several years of autopsies, therefore different protocols, which work equally well for the identification of PLP expression, were used. After placement on a heating plate (58 °C) for 30–60 min, sections were deparaffinized in a series of 100% xylene (3 × 10 min), 100% ethanol (2 × 5 min) and subsequently 5 min each in 96% ethanol, 80% ethanol, 70% ethanol and distilled water. Antigen retrieval was done by steaming the sections for 30 min in 10 mM Tris-EDTA buffer (pH 9.0; Promega, Leiden, the Netherlands). After rinsing in TBS (3 × 5 min), endogenous peroxidase activity was blocked with a 0.3% H2O2 in TBS solution (30 min). The sections were rinsed for 3 × 5 min with TBS. The primary anti-PLP monoclonal antibody (mouse anti-bovine for Myelin Proteolipid Protein (PLP); 1:500; MCA839G; BioRad, Hercules, CA, USA), diluted in a 3% BSA in TBS solution or in a 1% Normal Goat Serum in TBS-T, was incubated with the tissue overnight at 4 °C. For incubation a secondary biotinylated donkey anti-mouse antibody (diluted 1:400 in 3% BSA in TBS, for 2 h) or EnVision (Agilent Technologies, Santa Clara, CA, USA; lasted 30 min) was used. ABC incubation (diluted 1:400 in TBS) for 30 min was performed to create a complex with biotin. Subsequently, sections were incubated in DAB (for 5–10 min). Haematoxylin (1 min) was used for nuclear staining in all sections. After dehydration in successive ethanol 70%, 80%, 96%, 100% (2×) and xylene 100% (3×), sections were coverslipped with entellan.

### 4.3. Preliminary Screening

All of the LFB-stained sections were initially screened to establish the total number of tissue samples containing WM lesions. GM lesion types 2, 3 and 4 (which are limited to the GM only) were not visible on LFB-stained samples, due to the decolorization steps and therefore excluded from analysis [41]. Only GM type 1 lesions (covering both the WM and GM) were included, because they contain WM pathology that was assessable in the same manner as other included WM lesions. WM lesions were microscopically assessed for the presence remyelinating or demyelinating characteristics.

### 4.4. Microscopic Criteria for Demyelination, Remyelination and DAWM

Figure 4 represents a flowchart for in- or exclusion of WM pathology and for lesion classification. All tissue sections were systematically analyzed for the presence of WM pathology. Examples of normal myelinated fibers, DAWM, NAWM and demyelination are shown by their representative LFB staining (Appendix A). Abnormal WM areas with a clearly visible border were classified as WM lesions. When WM pathology did not contain a well-demarcated border (i.e., when there was diffuse myelin abnormality manifesting as a slightly diffuse, paler LFB-stained area), it was classified as DAWM. Subsequently, WM lesions were further classified as fully demyelinated or as having signs of remyelination. Full demyelination was present in lesions with a sharply defined border and no myelin within the border. When myelin was present within the border, lesions were classified as (partially) de- or remyelinated lesions according to the myelin characteristics. Characteristics of remyelination are paler blue LFB stained areas within lesions consisting of thinner and shorter myelin fibers that run either crisscross or, in shadow lesions, continuous to the fiber direction of the NAWM. Lesions showing signs of remyelination were divided into either partial or complete remyelination, based on previously established scoring criteria [19,20,22]. Partially remyelinated lesions show either (1) little remyelination that is restricted to the lesion rim and characterized by a paler gradient or (2) substantial remyelination, between 21–60% of lesion area remyelinated, either confluent or with patches of remyelination, and not necessarily restricted to the lesion rim (Figure 5). Lesions classified as (3) completely remyelinated show nearly complete remyelination, or >60% covering the entire lesion area, which is known as a shadow lesion (SL) [19,20,22].

### 4.5. Quantitative Analysis of Myelin

#### 4.5.1. Pre-Analysis Image Alterations

Figure 6 provides a complete overview of the quantification procedure. LFB stained histological sections were digitally scanned using a slide scanner (Vector Polaris, Perkin Elmer Waltham, MA, USA). Microphotographs were obtained with Phenochart (version 1.0.12, Perkin Elmer AKOYA Biosciences, Marlborough, MA, USA). Micrographs of LFB stained sections were taken at the highest possible magnification that allowed for the entire lesion to be visible. Three pictures of NAWM were taken with the same magnification as the lesion image. All images were then processed and analyzed using publicly available ImageJ software (version 2.0.0/1.52p, https://imagej.net/Fiji/, accessed on 4 June 2020). Firstly, all micrographs were processed into a grey-scale 8-bit image. Secondly, the image was made ready for thresholding into a binary black and white pixel image. For this, within the ‘Binary Options’ menu the ‘Black background’ option was checked to make sure that black pixels represented the myelin when a threshold value was set. Threshold values were set at a level that best matched the real blue intensity of the LFB image, in order to retain as many true pixels representing myelin as possible (Figure 6A,B). The contrast on the black and white image was visually equated with the corresponding LFB image. This resulted in a black and white binary image with either white pixels (value “255”) or black pixels (value “0”), with black being equivalent to myelin.

#### 4.5.2. Selection of Regions of Interest

In the black and white binary images, Regions-Of-Interest (ROIs) were manually drawn on the lesion border, on three different NAWM areas (Figure 6) and on the border of DAWM (Appendix A), excluding (large) blood vessels, tissue folds or other irregularities. When large vessels were present within the lesion, a separate ROI was drawn surrounding the respective vessels. The pixels of vessels were then subtracted from the total pixels in lesions (Figure 6).

#### 4.5.3. Myelin Quantity Analysis

The myelin quantity was analyzed in these ROIs by calculating pixel ratios. The total number of pixels (PT) and the number of black pixels (PB) within each ROI were measured. This was converted into myelin percentages (% black pixels) present within the ROIs ((PB/PT) × 100%). The percentage of myelin from three NAWM ROIs were averaged and used as standard maximum myelin density (as NAWM represents normal myelin density). For large blood vessels present within lesions, the total pixel count and black pixel count in these separately drawn ROIs were subtracted from the PT and PB in lesions. For this methodological research the myelin quantity of WM lesions was assessed by three independent raters to study the inter-rater reliability. Each rater applied a threshold value according to their own insight, adhering to the original LFB image. All three raters used the same ROIs. To assess if our method for myelin quantification can be further standardized and automated, the built-in automatic threshold option in ImageJ (‘Default’) was applied to WM lesions resulting in automatically generated myelin quantity. The same ROIs were used. Additionally, the sub analysis of a selection of images with DAWM and NWM of NNCs was done to investigate the method’s broader applicability. For ROIs of DAWM areas the percentage of myelin was determined in the same manner as described above.

### 4.6. Statistical Analysis

All statistical analyses were carried out in SPSS (version 22). Mean values and standard deviations (SD) of myelin quantity were computed for the lesions, NAWM and NWM. Mean myelin percentages of all three NAWM ROIs per case were calculated to correct for minor differences in normal myelin densities. Differences in myelin density between NAWM of MS patients and NWM of NNCs were compared using an unpaired *t*-test. The threshold for significance was set at *p*-value < 0.05.

The intra-class correlation coefficient (ICC) was calculated to determine the inter-rater reliability that reflects the agreement between the myelin quantification measurements. As the three raters were randomly selected and considered representative of a larger population of similar raters, a two-way random effects model was calculated (ICC (2,1)), with a 95% confidence interval (CI) [42]. ICC (2,1) was used to make findings of the three raters applicable beyond this study, as well as ICC-agreement, which considers systematic- and random errors [43]. Based on the 95% confidence interval of the ICC, an ICC of <0.50 was indicative of poor reliability, an ICC between 0.50 and 0.75 was indicative of moderate reliability, values between 0.75 and 0.90 indicated good reliability and an ICC >0.90 was indicative of an excellent reliability [44]. The ICC was calculated between the rates of 55 WM lesions from the three individual raters. The ICC was recalculated by using the built-in automatic threshold (default) setting in ImageJ as the fourth observer, which was then compared to rater 1. Bland-Altman plots were used to portray the agreement between raters for each subject against their mean [26]. Log-transformations were performed when indicated, which can be useful in stabilizing the variance caused by heterogeneous variation [45]. The Inter-rater reliability and Bland-Altman plots were reported employing standard recognized statistical techniques [26,42].

## Figures and Tables

**Figure 1 ijms-22-12634-f001:**
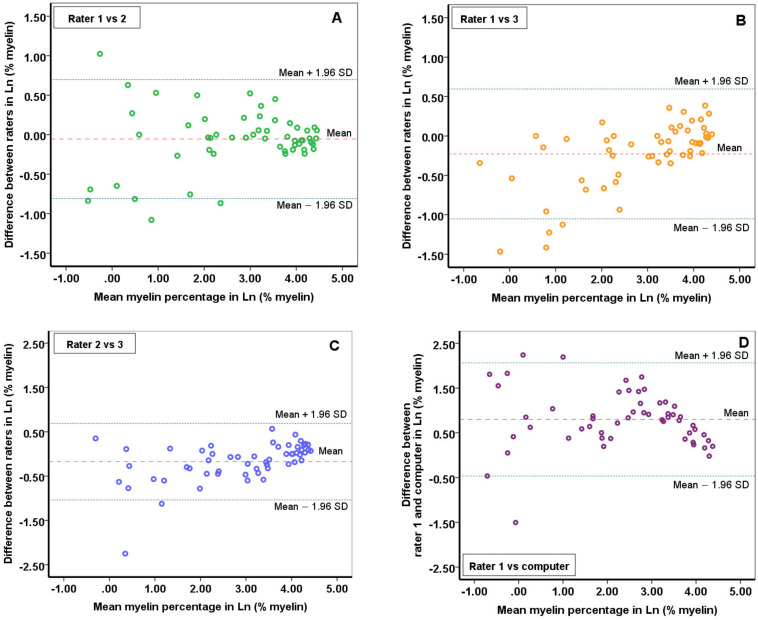
Bland-Altman plots showing inter-rater reliability between raters. (**A**) Showing inter-rater reliability between rater 1 and rater 2. It illustrates the mean difference (4.50% myelin) of measuring the myelin quantity, as well as the amount of scatter around the mean for each set of measurements (95% limit of agreement: −3.85 to 12.85%) after log transformation; (**B**) Showing inter-rater reliability between rater 1 and rater 3. It illustrates the mean difference (5.53%) of the measured myelin quantity, as well as the amount of scatter around the mean for each set of measurements (95% limit of agreement: −5.18 to 16.25%) after log transformation; (**C**) Showing inter-rater reliability between rater 2 and rater 3. It illustrates the mean difference (6.25%) of the measured myelin quantity, as well as the amount of scatter around the mean for each set of measurements (95% limit of agreement: −5.93 to 18.43%) after log transformation; (**D**) Showing inter-rater reliability between rater 1 and the default-method of ImageJ. It illustrates the mean difference (14.29%) of the measured myelin quantity, as well as the amount of scatter around the mean for each set of measurements (95% limit of agreement: −9.29 to 37.87%) after log transformation. Hence, the limit of agreements for the differences are in this figure illustrated in proportion to the mean (Ln).

**Figure 2 ijms-22-12634-f002:**
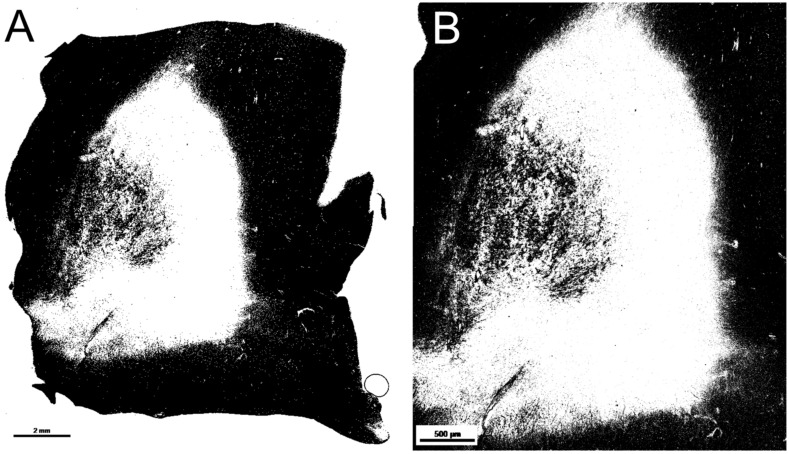
Remyelinated lesion (score 2) at different magnifications. (**A**) represents the lesion at magnification 0.6 × and (**B**) at magnification 1.9×. Ratio black pixels is provided in the sub analysis above (Table 1).

**Figure 3 ijms-22-12634-f003:**
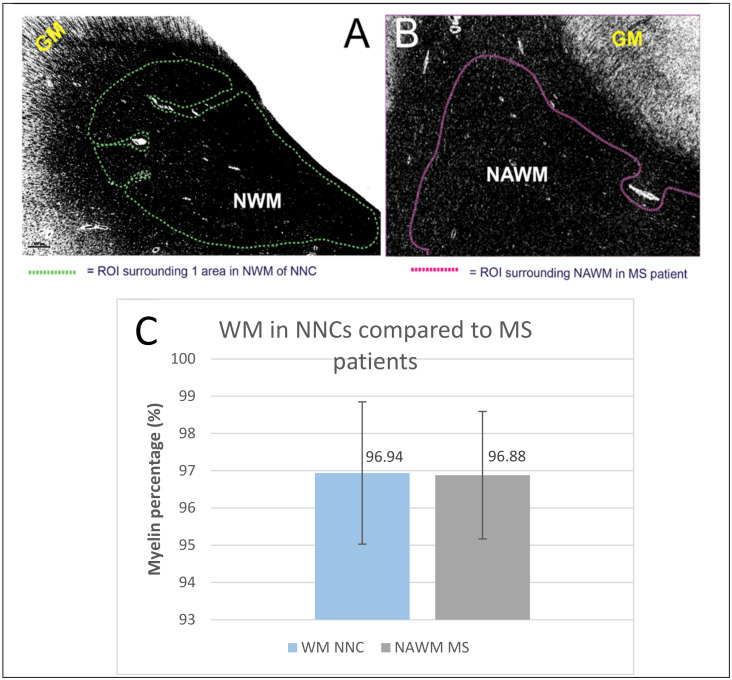
NAWM and NWM thresholded images for myelin density computation. (**A**) represents one of the three thresholded NWM images of a NNC brain section, illustrated with one ROI; (**B**) represents a thresholded image and ROI of NAWM in a MS brain section. LFB = luxol fast blue, NWM = normal white matter, GM = grey matter, NNC = non-neurological control, ROI = region of interest; (**C**) Bar graph representing the difference in WM quantity between NNCs (mean: 96.94%; SD 1.91%) and MS patients in the normal (appearing) white matter (mean: 96,88%; SD 1.71%). *p* = 0.949.

**Figure 4 ijms-22-12634-f004:**
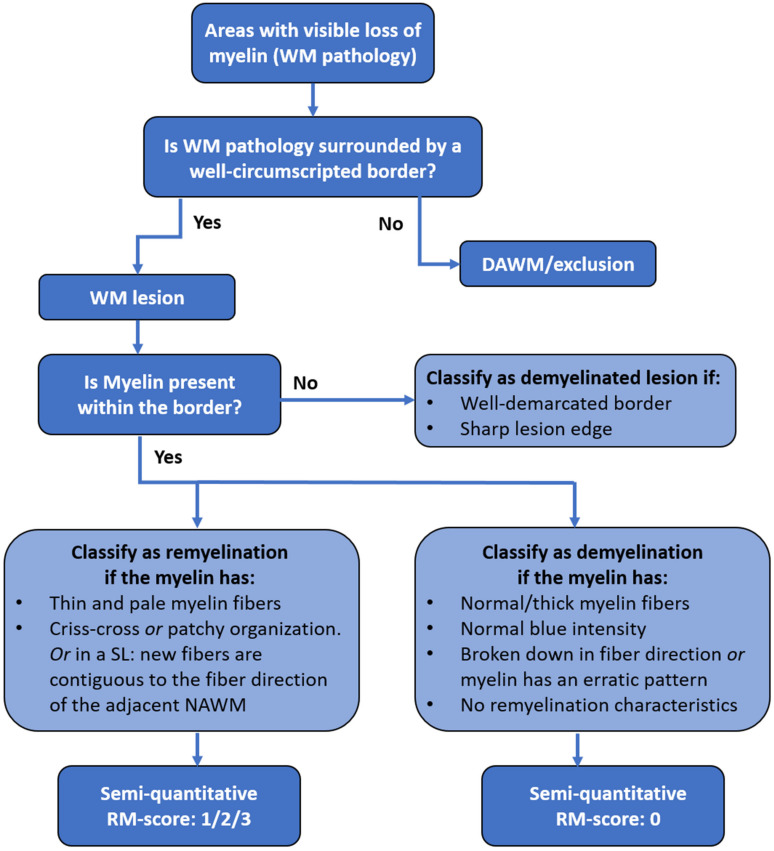
Flowchart for in- or exclusion of WM pathology and for classification of a lesion as a de- or remyelinating lesion. When WM pathology did not contain a well-demarcated border, it was classified as diffusely appearing WM (DAWM) and excluded from further analysis. The next step is to determine if there is myelin present within the border and to classify this as de- or remyelination using the semi-quantitative remyelination score. With (0) demyelination (0%, no remyelinating characteristics); (1) little remyelination (up to 20% restricted to the lesion rim); (2) substantial remyelination (21–60% of lesion area remyelinated, either confluent or patches of remyelination) and (3) nearly complete remyelination (>60%, covering the entire lesion area, SL) [19,20].

**Figure 5 ijms-22-12634-f005:**
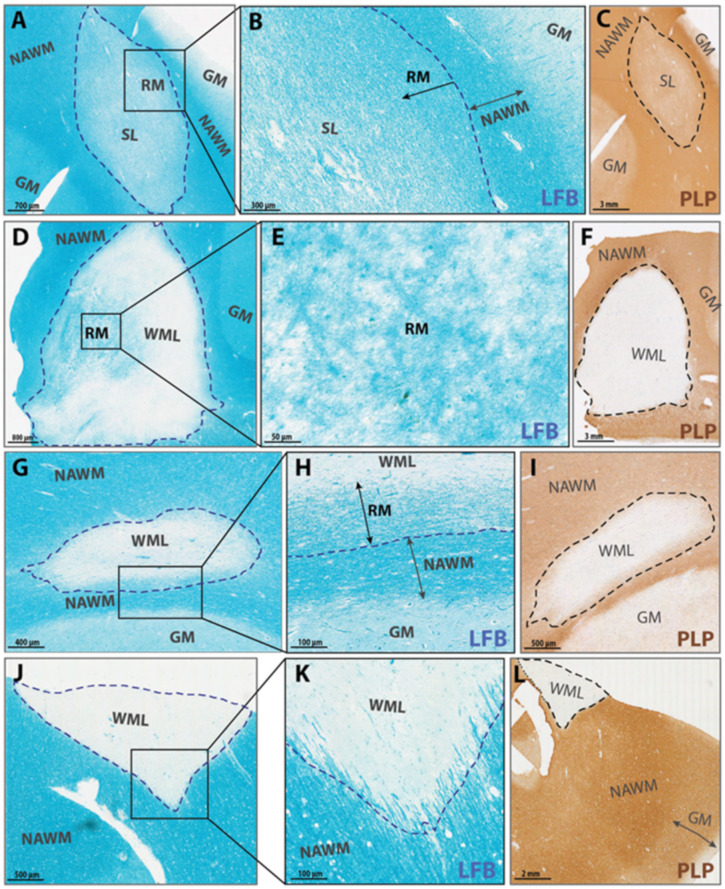
LFB (blue) stained WM Lesions with remyelination score 3, 2, 1, 0, respectively (**A**–**L**). [19,20,22] (**A**–**C**) represent a shadow lesion (SL, remyelination score 3). (**A**) showing an almost completely remyelinated SL, covering almost the entire surface, but still with lower myelin density in the center. Fibers are running almost similar to the fibers of adjacent NAWM. (**B**) The lesion border is still visible, with a visible gradient in myelin density (indicated by arrows). (**D**–**F**) Lesion with substantial remyelination (remyelination score 2), with thin, patchy myelin fibers spread throughout the lesion area (**E**). (**G**–**I**) Lesion with partial remyelination (remyelination score 1), showing a thinner myelin gradient restricted to the lesion rim (indicated by the arrow in (**H**)). (**J**–**L**) Demyelinated lesion (remyelination score 0) with at the border normal myelin thickness and cut in the same fiber direction as the surrounding NAWM. (**C**,**F**,**I**,**L**) Corresponding immunohistochemical myelin staining (PLP; brown) showing NAWM, WML, GM and SL.

**Figure 6 ijms-22-12634-f006:**
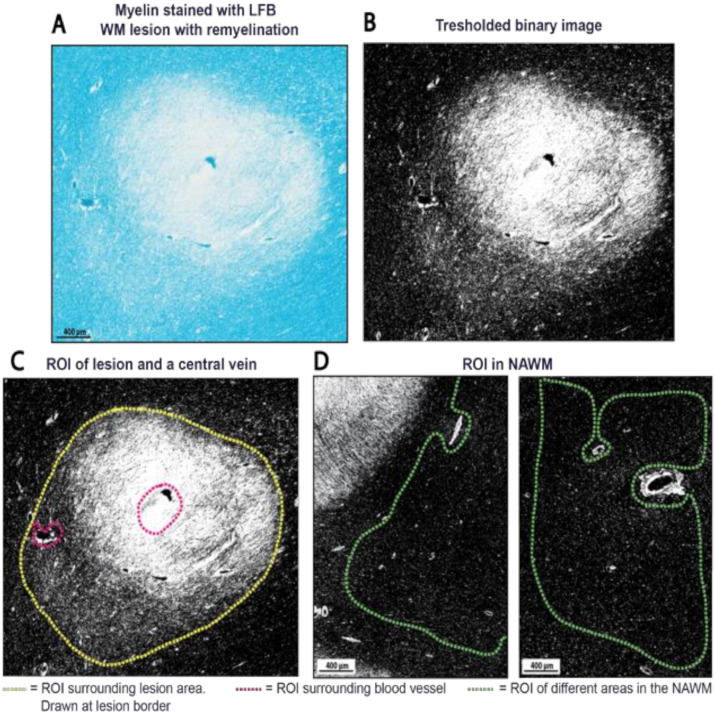
Myelin percentage computation with ImageJ (version 2.0.0/1.52p, https://imagej.net/Fiji/, accessed on 13 February 2020). (**A**) The percentage of myelin was quantified from images of Luxol Fast Blue (LFB) stained sections; (**B**) The images were processed into an 8-bit binarized mask of myelin, with the black pixels representing the myelin. A threshold was set at a level that most closely matched the true image (in LFB), in order to retain as many true myelin pixels as possible; (**C**) After ImageJ processing, the ratio of black pixels was divided by the total amount of pixels in the defined region of interest (ROI). The total pixel amount and black pixels of a blood vessel (present within the ROI) were subtracted from the total pixel amount and black pixel amount of the ROI to obtain only percentages of myelin; (**D**) In every brain section, three different ROIs were drawn in the NAWM. The images were taken at the same magnification as the lesion’s image magnification (to maintain the same pixel scale in every image) using Phenochart (version 1.0.12, Perkin Elmer AKOYA Biosciences). The ROI in the NAWM was drawn as pictured above, excluding (large) blood vessels. The percentage of black pixels was computed in the ROI for all 3 areas and averaged to a percentage of myelin. The number of black pixels then indicated the percentage of myelin in lesions, with 100% myelin being defined by the number of black pixels in NAWM (as NAWM represents normal myelin density).

**Table 1 ijms-22-12634-t001:** Quality analysis of lesion images taken at different magnifications.

Lesion with Partial Remyelination (Score 2)	Black Pixel Percentages ((PB/PT) × 100%)	Threshold Value
Analysis A. small magnification ^1^ Lesion: 0.6x	(475,272/2,158,826) × 100% = 21.24% black pixels within ROI	208
Analysis B. bigger magnification ^2^ Lesion: 1.9x	(44,856/211,200) × 100% = 22.02% black pixels within ROI	208

Analysis A and B are performed on the same lesion, but with a different image magnification. ^1^ Analysis A shows the ratio of black pixels (% black pixels) when the image magnification of the lesion was taken at 0.6×. ^2^ Analysis B shows a slightly higher ratio of black pixels when a higher lesion magnification was used. The same threshold was applied on both images. P^B^ = black pixel count, P^T^ = total pixel count, ROI = region of interest.

## Data Availability

Not applicable.

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
