# Peer review of "Myelin Quantification in White Matter Pathology of Progressive Multiple Sclerosis Post-Mortem Brain Samples: A New Approach for Quantifying Remyelination"

_ijms, 2021, doi:10.3390/ijms222312634_

Round 1
Reviewer 1 Report
In this study authors tried to report a new histopathological myelination quantification method. To gain this goal authors used Paraffin-embedded post-mortem MS brain tissue. For the assessment of myelin quantity, they used 14 MS patients and 5 non-neurological controls. The brain sections stained with Luxol Fast Blue to demonstrate shadow lesions and proteolipid protein to distinguish the WM-GM border and to identify areas of demyelination. Light microscopy and histological assessment used to check the demyelination and remyelination. The myelin quantity was estimated with a novel myelin quantification method in ImageJ. Finally, the authors concluded that by combining the novel quantification method in ImageJ with the microscopically examined remyelination criteria, the 280 degree of remyelination can be estimated more precisely and objectively in post-mortem brain tissue of MS patients.
Overall, this is interesting topic and research question. The manuscript is well written and adequate methods were employed to answer the question of the research.
There are some minor questions:
There are 32 tissue blocks from 14 MS cases and 5 tissue blocks from 5 non-neurological controls. How do you justify this difference in controls as well as tissue blocks numbers differences?
What is the application of this method in MS patients (alive) and is there any chance to use this method for therapeutic purposes? (Because here in this manuscript, mostly the histological techniques were used and there is no imaging (in alive humans) so that it could be useful also for patients). I would suggest discussing this in discussion section so that it can be more interesting for future studies in patients.
Have the authors tried to use animal models? In case of using the animal models in appropriate number, this method can be generalized and not only to be specific to post mortem samples.
Minor comments:
I would suggest using the dot after the citations in the text i.e. .[7,12,16,17] to [7,12,16,17].
There are minor typos in the text for example in line 164 : Using the LFB staining we observed and interesting phenomenon that confirms our recent observations.[25] is it and or an?
The figure 4. The chart picture quality is low and it can be improved.
Author Response
Response to Reviewer 1
Q1: There are 32 tissue blocks from 14 MS cases and 5 tissue blocks from 5 non-neurological controls. How do you justify this difference in controls as well as tissue blocks numbers differences?
A1: The pathological manifestation of MS is known to be very heterogeneous (e.g. Kuhlmann et al., 2017). Therefore a large number of patients and tissue blocks is required to obtain a representative sample, particularly when focusing on the process of remyelination, the extent of which can vary widely (as shown is this paper). Non-neurological controls on the other hand are expected to have a healthy, stable myelin density and as such fewer blocks are needed for a representative sample.
Q2: What is the application of this method in MS patients (alive) and is there any chance to use this method for therapeutic purposes? (Because here in this manuscript, mostly the histological techniques were used and there is no imaging (in alive humans) so that it could be useful also for patients). I would suggest discussing this in discussion section so that it can be more interesting for future studies in patients.
A2: The current study is indeed focused on the quantification of myelin in post mortem samples, but it may have clinical or therapeutic uses in the future. We agree with the reviewer that combining post mortem and clinical imaging data with the detailed analysis of myelin described here is a promising approach. We have added a sentence under ‘3.3. Conclusion and future prospects’ to include this idea: ‘Subsequent studies are needed to discover whether this method provides useful information of clinically relevant pathology. A potentially fruitful approach is combining (post mortem) imaging methods with the MQM.’ (line 300).
Q3: Have the authors tried to use animal models? In case of using the animal models in appropriate number, this method can be generalized and not only to be specific to post mortem samples.
A3: We here report on the application of our method in human brain; the inclusion of animal work is beyond the scope of the current work. Nevertheless, several animal models for de- and remyelination exist and these are often characterized by a more generalized and predictable level of pathology than human MS pathology is. In fact, we expect our myelin quantification method to be very well suited for the assessment of myelin in these animal models as well. We have modified the conclusion section so it now includes this suggestion (line 298).
Minor comments:
Q4: I would suggest using the dot after the citations in the text i.e. .[7,12,16,17] to [7,12,16,17].
A4: Either option is acceptable to the authors. We will leave the decision up to the editor/typesetter.
Q5: There are minor typos in the text for example in line 164 : Using the LFB staining we observed and interesting phenomenon that confirms our recent observations.[25] is it and or an?
A5: It should indeed be ‘an’. Text changed accordingly. (line 164)
Q6: The figure 4. The chart picture quality is low and it can be improved.
A6: An updated high resolution version of the image is now included in the manuscript. (page 12)

Reviewer 2 Report
In this study, the authors established a novel histopathological semi-quantitative assessment of remyelination with LFB staining followed by Image J software analysis. They defined criteria for a variety of myelination status in human post-mortem samples, like NNC, DAWM, NAWM etc. Although the authors proposed clear criteria, it should be noted that LFB can stain all myelinated fibers, not only remyelinated fibers. Also, histological staining, like LFB could be very variable. These factors will make the criteria in this paper potentially subjective. Finding a specific marker for remeylination would be much more reliable than LFB staining. My further questions and suggestion are listed below:
- Is it possible that the defined partial remyelination characterized by thin and pale myelin fibers, could also be in the process of demyelination rather than remyelination? How can you distinguish these two processes? In addition, how can you distinguish complete remyelination and normal myelination fibers?
- It’s necessary to provide a figure to make comparisons of normal myelinated fiber, DAWM, NAWM and demyelination by showing their representative LFB staining.
Author Response
Response to Reviewer 2
Q1: Although the authors proposed clear criteria, it should be noted that LFB can stain all myelinated fibers, not only remyelinated fibers. Also, histological staining, like LFB could be very variable. These factors will make the criteria in this paper potentially subjective. Finding a specific marker for remeylination would be much more reliable than LFB staining.
A1: We are aware that LFB is a general staining for myelin, not just remyelinated fibers. We acknowledge that the histological technique using LFB is subjective by its very nature, yet it is currently one of the best histological tools at our disposal. Working according to standardized procedures we therefore mention in the materials and methods section ‘All sections were washed and differentiated one by one.’ (line 334) and ‘The latter step (i.e. differentiation) was performed carefully and was checked microscopically to ensure homogeneity of the LFB staining. The differentiation steps were repeated if needed.’ (line 338).
We fully agree with the reviewer that finding a specific marker for remyelination would be ideal, but this is beyond the aims of our current paper. Mikael Simons’ group reported on BCAS1 as a marker for early myelinating oligodendrocytes that are found in a proportion of chronic white matter lesions of patients with multiple sclerosis (Fard et al., 2017). We have added reference to this work to the discussion section. It now reads: ‘With manual staining of fixed tissue sections, it is difficult to uniformly apply histological staining, which is an inherent disadvantage of all staining methods in post-mortem research. Established markers for remyelination are hard to find, although Fard et al., report on BCAS1 as a marker for early myelinating oligodendrocytes that are found in a proportion of chronic white matter lesions of patients with MS.’ (line 224).
My further questions and suggestion are listed below:
Q2: Is it possible that the defined partial remyelination characterized by thin and pale myelin fibers, could also be in the process of demyelination rather than remyelination? How can you distinguish these two processes?
A2: This is a fundamental issue. We here assume that myelin loss is caused by a destructive process and that remeyelination by oligodendrocytes producing new myelin that enwraps the axon initially as a thin layer, is making demyelination to look quite different from remyelination. This leaves open the possibility that during demyelination myelin sheets are retracted by pressured oligodendrocytes in a subtle and orchestrated manner, making them look similar to the thin layer of myelin typically observed with remyelination. Even though it is currently unknown if this subtle retraction of myelin happens at all in MS, this is a limitation of our study, since we cannot distinguish these two processes. We now point this out as a limitation in the discussion section (line 254).
Q3: In addition, how can you distinguish complete remyelination and normal myelination fibers?
A3: It is impossible to distinguish complete remyelination and normal full myelination by using the tissue and methods described here. We now make mention of this under ‘3.2. Study strengths and limitations’. (line 262)
Q4: It’s necessary to provide a figure to make comparisons of normal myelinated fiber, DAWM, NAWM and demyelination by showing their representative LFB staining.
A4: The paper now includes a figure (supplementary figure B4) that reflects the representative LFB staining of DAWM, NAWM and demyelination, referred to in the materials and methods section (line 375).
